# Localization of Scattering Objects Using Neural Networks

**DOI:** 10.3390/s21010011

**Published:** 2020-12-22

**Authors:** Domonkos Haffner, Ferenc Izsák

**Affiliations:** 1Institute of Physics, Eötvös Loránd University, Pázmány P. stny. 1A, 1117 Budapest, Hungary; s24q2w@caesar.elte.hu; 2Department of Applied Analysis and Computational Mathematics & NumNet MTA-ELTE Research Group, Eötvös Loránd University, Pázmány P. stny. 1C, 1117 Budapest, Hungary

**Keywords:** localization problem, inverse scattering, neural networks, scattered data

## Abstract

The localization of multiple scattering objects is performed while using scattered waves. An up-to-date approach: neural networks are used to estimate the corresponding locations. In the scattering phenomenon under investigation, we assume known incident plane waves, fully reflecting balls with known diameters and measurement data of the scattered wave on one fixed segment. The training data are constructed while using the simulation package μ-diff in Matlab. The structure of the neural networks, which are widely used for similar purposes, is further developed. A complex locally connected layer is the main compound of the proposed setup. With this and an appropriate preprocessing of the training data set, the number of parameters can be kept at a relatively low level. As a result, using a relatively large training data set, the unknown locations of the objects can be estimated effectively.

## 1. Introduction

Determining the location and material properties of objects is an important practical problem in a number of measurement systems. For this, a general approach is to detect how the objects scatter certain incoming waves. One can apply either sonic or electromagnetic waves, depending on the experimental setup. This general framework includes radar and sonar systems, seismic methods in geophysics, and a number of methods in medical imaging, such as ultrasonic methods and computed tomography. In many cases, only the reflection of certain pulse-waves is measured. At the same time, a scattered periodic wave involves more information, which can make it possible to perform the quantitative analysis of the scattered objects. This is used, e.g., in ground penetrating radar systems and through-the-wall imaging for monitoring buildings, assisting archaeological explorations and seismic surveys. In this case, given point-source wave pulses or plane waves are emitted. Typically, time-harmonic waves are used, often with multiple frequencies. In realistic cases, even the detection of the scattered waves is non-trivial, since one can only detect these in a very limited region in contrast to a conventional computer tomography or magnetic resonance imaging.

The corresponding mathematical models—leading to inverse problems—contain a number of challenges also for the theoretical studies. Accordingly, different numerical methods were developed in order to assist and fine-tune different radar systems, ultra sound systems or evaluate various geological measurements. All of these conventional methods use a kind of regularization, since the “raw” mathematical equations corresponding to the physical laws lead to unstable and ill-posed problems. Even these regularized methods, which become stable and possess unique solutions, are computationally rather expensive and somewhat arbitrary, owing to the choice of the regularization. In any case, these lead to involved models and the corresponding simulations demand extensive computational resources.

Neural networks offer a modern approach for replacing this conventional method [1,2]. Having a large number of direct measurements or simulations, one can try to automatize the solution and develop a relatively easy localization procedure using neural networks. In this case, the time-consuming compound can be to obtain sufficiently large training data and train the network. This can be what identified with an immense number of observations and, based on this, carrying out a long calibration procedure. Having a trained network at hand, the corresponding inverse problem can be solved quickly, even in real-time.

Accordingly, in the last decade, a number of neural networks were developed for specific tasks to assist or replace conventional sensing and measuring.

### 1.1. Statement of Problem

We focus on the case, where a scattered plane wave is analyzed for finding the location of multiple objects. In order to mimic a realistic situation, the scattered waves were only detected in a limited region. In concrete terms

Two uniform unit disks with reflecting boundaries were placed in free space.Given plane waves were applied from several directions.The scattered waves were measured only on the bottom edge of a square-shaped domain, where the phenomenon was simulated.The position of the obstacles had to be determined while using the scattered waves.

Figure 1 and Figure 2 show the corresponding geometric setup and a sample scattered wave.

#### 1.1.1. Mathematical Model

The conventional direct mathematical model of wave scattering is the following partial differential equation (PDE from now on) for the unknown complex-valued function $u:R^{2}\to C$: (1)$$\left\{\begin{array}{c} \left(\Delta +k^{2}\right)u\left(x\right)=0,\;\;\mathrm{for}\;x\in R^{2}\backslash \bar{\Omega^{-}}\\ -\partial_{\nu (x)}u\left(x\right)=-\partial_{\nu (x)}u^{inc},\;\;\;\;\mathrm{for}\;x\in \partial \Omega^{-},\\ \lim_{|x|\to \infty }{\left|x\right|}^{\frac{d-1}{2}}\left(\partial_{r}u\left(x\right)-iku\left(x\right)\right)\;=\;0, \end{array}\right.$$
where |u| corresponds to the amplitude, $\Omega^{-}$ denotes (all of) the obstacles (the union of the two disks in Figure 1) with the surface $\partial \Omega^{-}$, and *k* denotes the frequency of the scattered wave. In the first line, the Helmholtz equation is given for the propagation, the boundary condition in the second line corresponds to the reflecting obstacles, while, in the third line, the usual Sommerfeld far-field radiation condition is formulated. For more details, we refer to the review paper [3].

For fixed obstacles $\Omega^{-}$, this leads to a well-posed problem, which is usually solved with an integral representation while using the Green function for the operator $\Delta +k^{2}$. This is also applied to perform training data for our approach.

However, we investigate an inverse problem for the following: the solution *u* is known in $\Omega_{0}\subset R^{2}\backslash \bar{\Omega^{-}}$ and we have to determine $\Omega^{-}$. For our case, $\Omega_{0}$ is neither an open subset of $R^{2}\backslash \bar{\Omega^{-}}$ nor a closed boundary: this is only a side of the computational domain. In this way, any theory ensuring the well-posedness cannot be applied for our inverse problem. Nevertheless, despite all of these difficulties, methods that are based on the mathematical analysis are developed, see, e.g., [4].

#### 1.1.2. Neural Network Approach

As mentioned, a recent research direction for investigating inverse problems that correspond to (Equation 1) is given by neural networks. Here, using an enormous number of location-scattered wave “pairs”, the network will learn to associate a pair of locations to a certain scattered wave.

A comprehensive work explaining the background, the basics, and dealing with earlier achievements can be found in [5,6]. We refer to these works in order to understand the standard structure of the corresponding neural networks.

In the last few years, these methods were developed further and applied to real-life cases: deep neural networks were constructed to also detect complex shapes using scattered waves in [7,8], and the method is also applicable in the case of non-linear waves [9]. A recent review on this topic with a number of further references can be found in [10]. Our geometrical setup and the equations in (Equation 1) are also related to the problem of sound source localization, which is also investigated while using neural networks, see, e.g., [11,12]. Detecting the structure of entire domains is one of the most important and challenging problems in this area, which have also been recently tackled by applying neural networks [13].

Obviously, the structure of the neural network has a significant impact on the results. The starting point of such constructions is mostly a well-working neural network structure, which is often used for image recognition problems. Such general structures for the present purposes are shown in [5,6].

In [14], a new direction considering the neural networks is suggested: efficient shallow networks are constructed, combined with special activation functions and parallel backpropagation gradient algorithms in order to keep the number of unknown parameters at a relatively low level.

The approach of the work is the same here: using a neural network with a special structure, involving a moderate number of parameters. Besides speeding up the corresponding simulations, this can prevent overfitting and lead to stable and reliable computing processes.

Another novelty of our approach is that scattered waves are only detected here on a narrow section, such that we use only a part of the information that arises from the scattering. Of course, this realistic case has its own limitation: complex structures can hardly be recognized in this way.

Recent results that are closely related to our work can be found in [7,8]. Here, the authors developed neural networks to detect very complex shapes. At the same time, they only assumed one object at a fixed position. Moreover, the scattered wave was detected all around the scattering object. We did not apply these assumptions, at the same time, only the locations of simple disks were detected.

## 2. Materials & Methods

### 2.1. Obtaining Training and Validation Data

In our case, the training data consist of a set of vector-pairs ${\left(F_{j},\;G_{j}\right)}_{j=1,2,\ldots ,J}$. Here $G_{j}=\left(G_{j1},G_{j2}\right)\in R^{\times }R^{2}$ corresponding to a geometrical setup, which can be identified with the two coordinate pairs of the disk-midpoints, while $F_{j}\in R^{161}$ denotes the dimensionless wave intensity in our observation points, as shown in Figure 3. Shortly and formally: the further task is to predict $G_{j1}$ and $G_{j2}$ while using $F_{j}$.

In order to obtain a satisfactory set of training data for this, we considered each geometrical setup of disks with different integer midpoint-coordinates, i.e., the elements of the set
$$G=\left\{\left(G_{j1},G_{j2}\right):G_{j1},G_{j2}\in {\left\{-4,-3,\ldots ,3,4\right\}}^{2},\;G_{j1}≻G_{j2}\right\}$$
where ≻ denotes the lexicographical ordering. This means 735 different cases, which proved to be sufficiently large. Note that, taking much more training data could not further increase the accuracy of our prediction. On the contrary, several runs made the learning process have a more oscillating accuracy.

For the pairs of disks that are given by G, we have simulated the scattered waves while using the simulation package μ-diff in Matlab [15]. In this framework, it is also possible to choose different boundary conditions and point sources for the incident waves. Figure 1 and Figure 2 depict sample simulated waves.

The dimensionless wave intensity is measured on the bottom edge of the computational domain in 161 gridpoints. For a geometrical setup $G_{j}$, this gives $F_{j}$. The choice of 161 is a good balance: this means approximately eight measurement points per the dimensionless wave length $\frac{1}{2}$, which is a minimum for identifying a scattered wave. Using more gridpoints could lead to extremely long simulation time and an unnecessarily large number of parameters.

A total of 92 percent of the generated data set was used for training, and the remaining eight percent for validation purposes. In concrete terms, we used J=676 samples for training and a validation data set with 59 samples, which were chosen randomly.

We have used sixteen different plane waves with angles: $k\frac{\pi }{16},\;k=0,1,\ldots ,15$ and a dimensionless wave length of $\frac{1}{2}$.

To summarize, the raw training data that we have worked with can be given by an array of size 16×676×161.

#### Preprocessing of Data

The data coming from the simulations can be highly oscillating, such that a small shift results in a completely different data at a fixed observation point. To get rid of this main problem, we considered observation points with maximal amplitude and the measurement was then interpolated and extrapolated from these locations, in order to obtain an estimated amplitude function in each 161 points of the measurement.

The corresponding process was also executed while using the built-in subroutines in Matlab. Figure 3 shows this interpolation step in a sample case.

One may expect that the simplified data set can have a shorter representation. Accordingly, we applied a max pooling layer before sending data to the convolution layer in order to push down the number of parameters in the consecutive computation.

It turned out that this radical simplification does not harm the prediction: we could successfully localize the disks while using 11 data points instead of the original number 161. One can observe in Figure 3 that the real simplification occurs in the previous step of the preprocessing.

All of this can lead to a loss of data. From a practical point of view, we may obtain then very similar transformed data sets that correspond to a completely different geometric setup of the scattering objects. This problem will also be discussed later.

This last step could also correspond to a layer within the neural network, but, in this discussion, we consider the above preprocessed data as the input of the network.

Additionally, by applying this procedure, noise can be filtered out, which pollutes most real observations. For testing purposes, we have added Gaussian noise to all of of our training and simulation data. The amplitude of the noise was one-fifth of the original plane waves’ standard deviation. The top of Figure 4 shows a pair of these. Additionally, in this example, one can compare the preprocessed data that arise from the original simulated data and the corresponding noisy in the bottom of Figure 4. Note that no additional denoising procedure was applied, and our preprocessing performed this task automatically.

### 2.2. The Structure of the Neural Network

When working with neural networks, the main challenge one can face is to figure out the structure to use. This usually consists of modifying an existing network and then fine-tuning the parameters that determine its structure. In concrete terms, one can modify the size of dense layers and convolution windows, the number of features in convolutions to obtain an optimal performance. Additionally, inserting dropout steps and tuning the parameters of the underlying optimization process is of great importance. This is a really time-consuming process.

As a starting point, we have used the neural networks shown in [12,16] and the general framework in [5]. Here, the data are immediately sent to consecutive convolutional layers and then the information is collected in one or more dense layers, which is finally summarized into the desired output values.

We present here one specific network, which has delivered the most accurate estimation for the locations of the unknown disks.

Finding appropriate convolution windows is an important building block in neural networks. In concrete terms, a convolution window $w=[w_{1},w_{2},\ldots ,w_{n}]$ transforms the vector $v=[v_{1},v_{2},\ldots ,v_{N}]$ with N>n to $w*v\in R^{N-n+1}$, as given by
(2)$$w*v=\left[\sum_{j=1}^{n}w_{j}v_{j},\sum_{j=1}^{n}w_{j}v_{j+1},\ldots ,\sum_{j=1}^{n}w_{j}v_{N-n+j}\right].$$

This window is associated with some “feature”, which is found by the neural network during the learning process by optimizing the components (or weights) of w. Usually, a couple of such convolution windows are included in the networks to gain all of the characteristic features in the training set.

The main improvement in our construction is that we have used a so-called two-dimensional locally connected layer. In this case, for all components in (Equation 2), we had to use different weights. Formally, the convolution window now becomes
$$w_{\mathrm{LC}}=\left[w_{1,1},w_{1,2},\ldots ,w_{1,n},w_{2,1},w_{2,2},\ldots ,w_{2,n},\ldots \ldots ,w_{N-n+1,1},w_{N-n+1,2},\ldots ,w_{N-n+1,n}\right]$$
and instead of (Equation 2), we have
(3)$$w_{\mathrm{LC}}*v=\left[\sum_{j=1}^{n}w_{1,j}v_{j},\sum_{j=1}^{n}w_{2,j}v_{j+1},\ldots ,\sum_{j=1}^{n}w_{N-n+1,j}v_{N-n+j}\right].$$

We defined a couple of these complicated windows in order to obtain the features of the scattered wave. Moreover, these are varied according to the angle of the incident plane waves. The motivation of this is that the observation points are located on the bottom edge and, therefore, depending on the angle of the incident plane waves, different kinds of scattering occur.

The description in the forthcoming points can be easily followed in Figure 5, where the overall structure of our network is demonstrated. The connections between the layers are represented with dashed and straight lines, according to their operations. When information was just passed through the layers (no parameters were added), we used dashed lines. In other cases, continuous lines correspond to operations with some parameters. For a clear visualization, the lengths of the layers are scaled down, so that they fit in one figure. The size (dimension) of the different layers is always displayed.

#### 2.2.1. the Layers

The first layer collects the information from the scattered waves that arise from given incident plane waves with 16 different angles. In concrete terms, after preprocessing, for each geometric setup, we have an input of size 11×16.

In the first hidden layer, for each incident wave direction, we allowed 12 different two-dimensional locally connected convolution windows of length 8×4.

The application of each two-dimensional convolution window in (Equation 3) to the input vectors of length 11, results in a vector of length 12−4+1=8. An arrow in Figure 5 corresponds to one of the convolutions in (Equation 3). This is performed in 12 cases, such that we obtain a matrix of size 8 × 12. The total size of the first hidden layer is 16 × 8 × 12 because we have 16 incident waves.

In the second hidden layer, we collect the information that is given by the first hidden layer. This does not increase the complexity of the model, since we did not use any weight or unknown parameters for this layer. This results in a layer that consists of one vector of length 1536.

The following hidden layers are all dense. In these layers, each component is affected by the entire data in the previous layer. In practice, dense layers are used when the information in the data of the previous layer has to be compressed into a smaller one. In our neural network, we achieved this by using dense layers with the lengths of 50, 40, and 8, respectively. Neither the numbers nor the sizes of these layers are indicated a priori. This choice is the result of a series of experiments.

Obviously, as an output, we have used a dense layer of size 4, which means the four coordinates of the unknown midpoints of the scattering objects.

#### 2.2.2. Parameter Reduction

The transformation between the layers is governed by parameters that, when combined with the above structure, completely describes the action of the neural network.

In concrete terms, we have used the following number of parameters:[(16×8)×(4×1)+(16×8×1)]×12+(16×12×8+1)×50+(50+1)×40+(40+1)×8+(8+1)×4=86,934.

Even though it seems to be an overly large number of parameters, in the practice of neural networks, this is still a moderate value. Observe that the major compound of this sum corresponds to the application of the first dense layer. The contribution of the second one is still larger than that of the locally connected layer.

The so-called overfitting effect is the real danger of using an unnecessary large number of parameters. When overfitting occurs, the prediction becomes consistently worse after a certain number of iterations. Therefore, “unnecessary” parameters should be eliminated. This is done automatically while using a dropout step [17] in some given layers. As it turned out, the optimal choice was a radical cut of the parameters by halving them in the first dense layer, such that, indeed, we used 48509 of them. Note that this step may also result in a loss of information. Accordingly, the application of further dropout steps led to less accurate predictions.

#### 2.2.3. The Activation Function

The data that we are using are bounded and positive. Therefore, we applied the RELU activation function in each step, but the final one. This real function is constant 0 for negative inputs and identical for positive ones. Other activation functions were also tried, but they delivered less accurate results. In the literature, for similar neural networks [18], a sigmoid activation function was applied in the last step. Our experiments confirm that this choice is optimal. Note that the complexity and the computational time is not affected by trying different activation functions.

### 2.3. Loss Function and Optimization

When working with neural networks, the difference between the predicted and real values must be minimized. We chose the mean squared error as a common measure of the prediction error. The accuracy of the neural network is measured by this value. In practice, we can only minimize the loss function of the training set and validate the result on the validation data set. A significant difference between the training and validation loss indicates the presence of overfitting. In this case, the model will be extremely inaccurate for everything, except the training dataset. To sum up, the quality of our predictions can be characterized with the validation loss.

The conventional optimization processes in neural networks are stochastic gradient methods [19], from which we chose the ADAM algorithm [20,21,22]. Because inverse problems are extremely unstable, a small change in the parameters can result in a completely different output. Accordingly, by the calibration of the parameters, a relatively small learning rate 0.0005 was the optimal one. Two further parameters had to be chosen in the ADAM algorithm; we took $\beta_{1}=0.9$ and $\beta_{2}=0.8$.

The number of global iteration steps, called the epochs, where the full training data set is used, was also experimentally determined. The process was terminated when the validation loss does not decrease any more.

## 3. Results

The neural network was implemented while using the Keras library in Python. Using the parameters shown above, after around 100 epochs, the validation loss function does not decrease significantly anymore. Therefore, taking a lot of further epochs would not increase the quality of our prediction. The train and validation loss are strongly correlated and they remain close to each other. This is a common indicator of the success of the learning process.

The oscillation of the loss function is accompanied with the optimization algorithm. The validation dataset is smaller and it was not used in the minimization procedure, such that it exhibits larger oscillations. Figure 6 shows this behavior, which is typical for neural networks.

### 3.1. Prediction of the Locations

In order to measure the quality of our prediction, we have computed the average squared error of the obstacle-midpoints. In Figure 6, one can observe that this value goes below 0.002 during the optimization process.

Figure 7 and Figure 8 display some concrete midpoint-predictions.

The larger disks represent the real size of the scattering objects, while the smaller show their midpoints in order to demonstrate the accuracy of the predictions. The deviations of the predicted and real midpoints are shown next to the graphs. The total (real) distance between the centers is given shortly by *dist*. We have used the short notations $dev_{b}$ and $dev_{g}$ for the values of deviations between the predicted and real centers, while *b* and *g* refer to the blue− and the green centers, respectively. These are summed up to obtain the variable dev_total.

We have calculated the distances between the scattered waves on the bottom edge in order to analyze the simulation results further. According to our previous optimization procedure, the distance was taken the square distance of the two data (of length 161). In this way, we intended to detect the problematic cases, where scattered waves are very close to each other. Figure 9 shows the distance distribution of the waves. As one can realize, a lot of relatively small distances occur, which clearly indicates the unstable nature of the present scattering problem. One would expect that the almost identical scattering waves correspond to the very similar geometry of the scattering objects. Surprisingly, this is not the case, as shown in Figure 10 and Figure 11.

A natural attempt to enhance the efficiency of the neural network is to remove these pairs from the training data set. Interestingly, this did not significantly improve the accuracy, pointing again to the rather complex nature of the problem.

We have also investigated the effect of noisy data in the simulations. Because our preprocessing could filter the additive noise, the input of the neural network was very slightly affected by this, see the bottom of Figure 4. Therefore, we obtained the same accuracy for the midpoint-prediction. We could not even separate the final results arising from noisy and noiseless data sets, due to the randomness of the splitting into training and learning data and the built-in stochastic optimization method ADAM.

### 3.2. Comparison with Other Approaches

The main compounds of our construction were the applications of an appropriate preprocessing of data and an appropriate locally connected layer in the network. In order to demonstrate the importance of these, we performed two additional series of simulations. For testing purposes, we have used noiseless data in all cases.

In the first series, we did not apply any preprocessing. In this case, the loss function remained 2–3 times the one in the original network, as shown in Figure 12. At the same time, since we had an input of size 16 × 161 (instead of 16 × 11), the computational time was about ten times more when compared to the original case.

In the second case, we have substituted the locally connected layer with a convolution layer, while using convolution windows. Here, the computational costs are at the same level as compared with the original case. At the same time, the loss function becomes about two times of the original value. Figure 13 shows the simulation results.

The following observations also refer to the power of our approach:Our network could process even the oscillating full data set with an acceptable loss.Using a conventional convolution network, train loss is above the validation loss, which suggests that new variables should be included. The locally connected layer purpose.

The simulation codes with detailed explanations, comments, and figures on the results can be found in the Appendix A. They are given as Python notebooks, where each of the main steps can be run independently. All of the relevant information on these is collected in the file readme.txt. It is completed with the data sets that were used for our simulation.

### 3.3. Computing Details

In order to point out the computational efficiency of our method, we mention that a simple laptop with Intel i3 processor and 4 GB RAM was used for the simulations.

In the following, we summarize the computing time of the main steps in the simulations.

Computing time of the simulation data: approx. 20 h.Computing time for the data transformation: approx. 10 s.Training the neural network and computing the prediction: 2–3 min.

## 4. Conclusions

A neural network based approach was presented for inverse scattering problems with restricted observation of the scattered data. Including a complex locally connected layer in the network seems to be the right balance: this ensures sufficient complexity, while a moderate number of parameters are used. A feasible preprocessing of the data was the other cornerstone: we have assisted the network by pre-mining information. In this way, the number of parameters could be reduced. This not only resulted in the speed of the simulations, but also made them more reliable by avoiding overfitting. Moreover, it has the capability to diminish the effect of noise.

This study provides a general framework, fixing that deep neural networks with an appropriate preprocessing and locally connected layers are perfect for this job.

At the same time, the present work has its limitations: in a robust method, the number and shape of the scattering objects and their material properties are unknown. For a corresponding extension, one should first detect the number of objects with a simple network and perform an enormous number of training data for different shapes and locations.

## Figures and Tables

**Figure 1 sensors-21-00011-f001:**
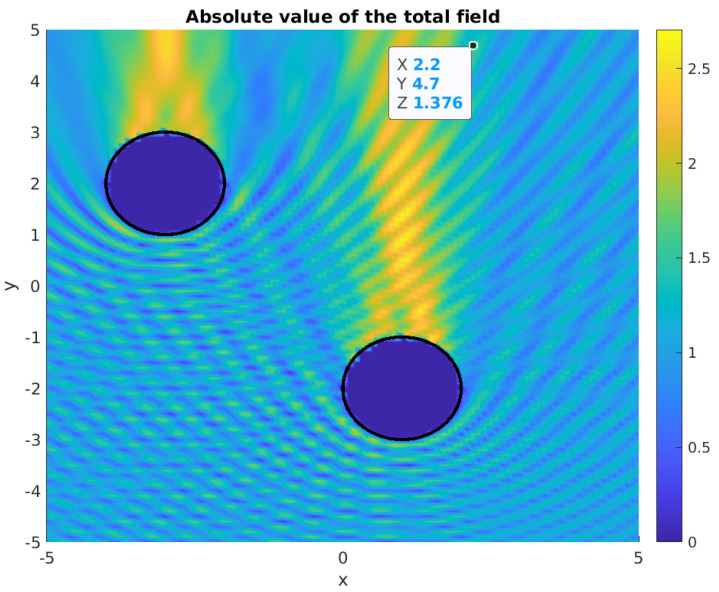
Absolute value of a full wave in a simulation with given obstacles. An upward planar incident wave was applied with k=4π.

**Figure 2 sensors-21-00011-f002:**
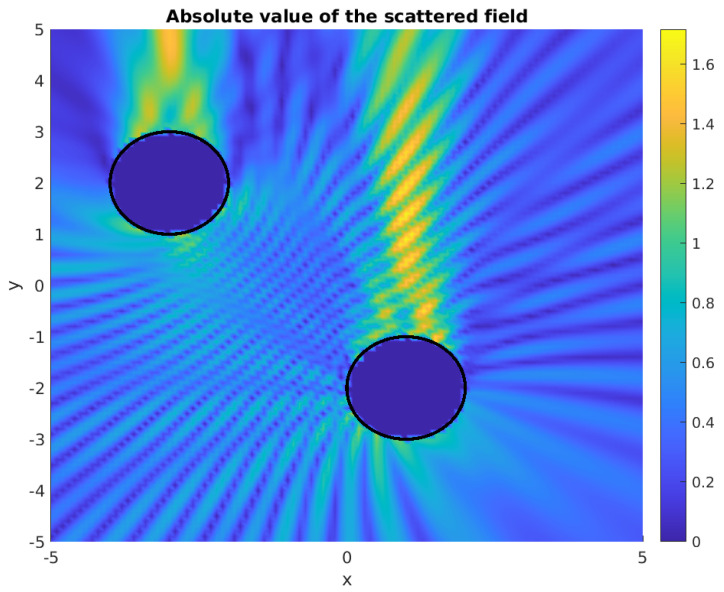
Absolute value of the scattered wave in a simulation with given obstacles. An upward planar incident wave was applied with k=4π.

**Figure 3 sensors-21-00011-f003:**
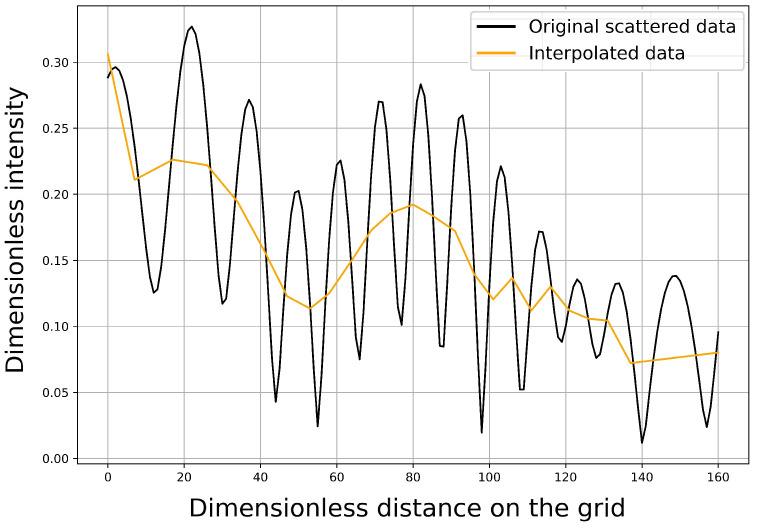
Sample row simulation data (without noise) and the interpolated data.

**Figure 4 sensors-21-00011-f004:**
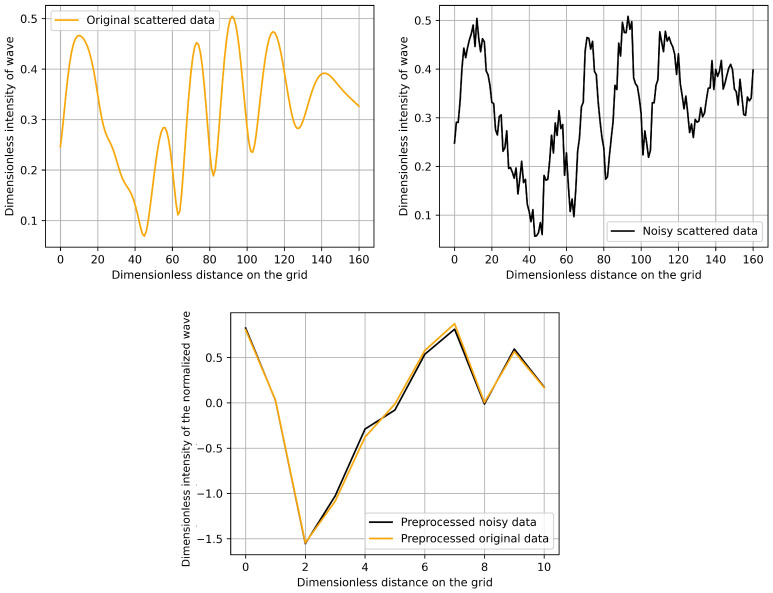
A sample simulated scattered data (**top left**) and a corresponding noisy data (**top right**) together with the preprocessed version of these data sets (**bottom**).

**Figure 5 sensors-21-00011-f005:**
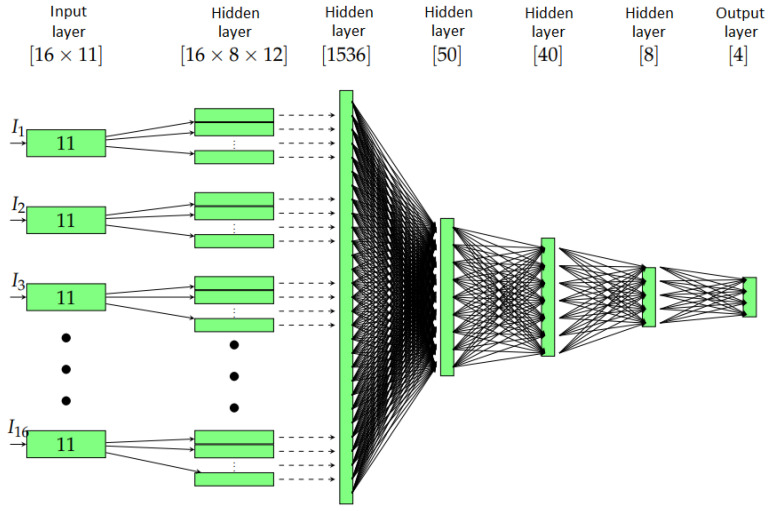
The structure of the neural network with the size of the consecutive layers and the inputs $I_{1},I_{2},\ldots ,I_{16}$.

**Figure 6 sensors-21-00011-f006:**
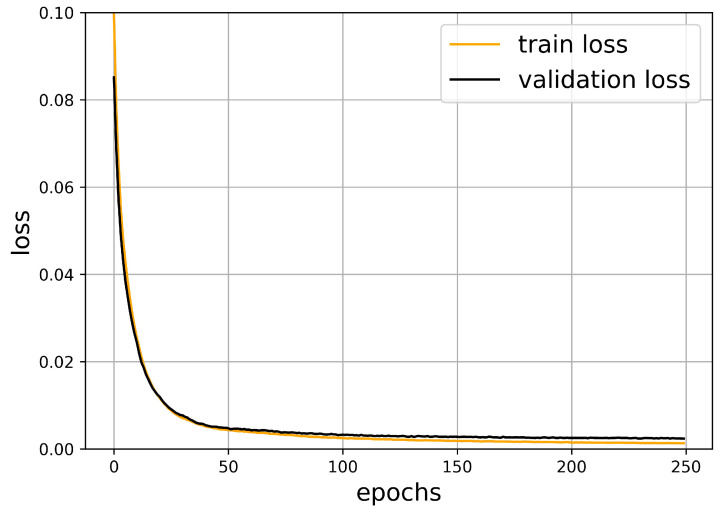
The train and validation loss in the function of epochs.

**Figure 7 sensors-21-00011-f007:**
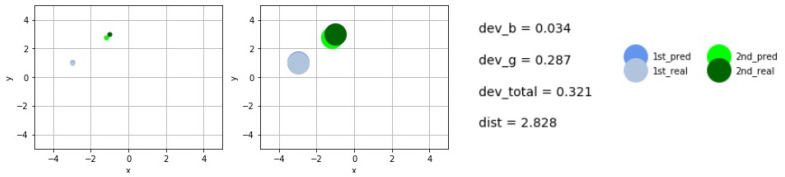
Example for the real and predicted midpoints and objects.

**Figure 8 sensors-21-00011-f008:**
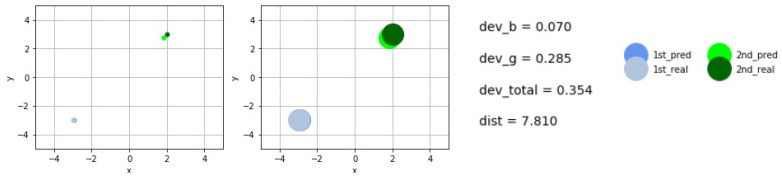
Example for the real and predicted midpoints and objects.

**Figure 9 sensors-21-00011-f009:**
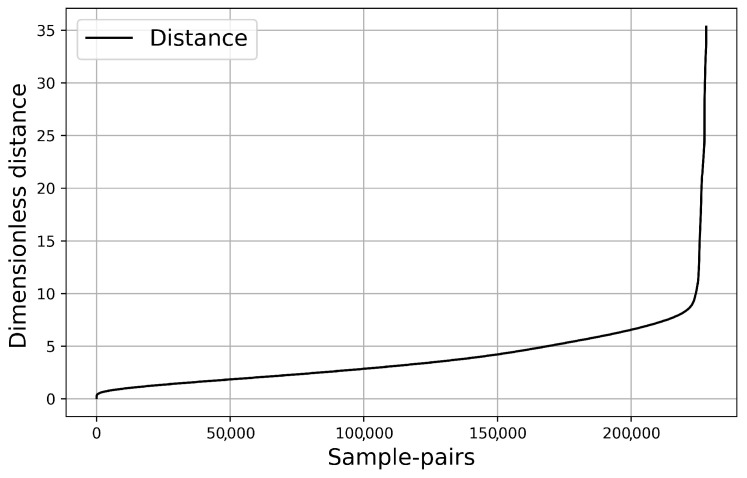
The distance distribution of the waves. *x* axis: the number of sample-pairs with less distance than *y*.

**Figure 10 sensors-21-00011-f010:**
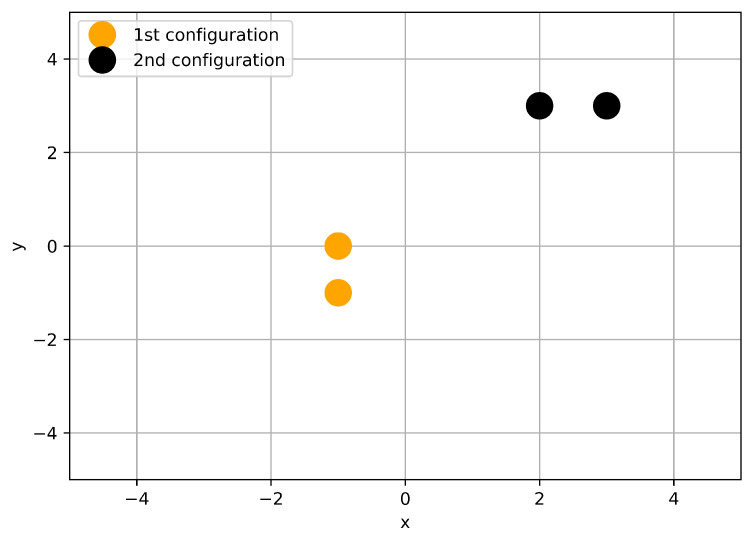
Two different geometric setup of the scattering obstacles.

**Figure 11 sensors-21-00011-f011:**
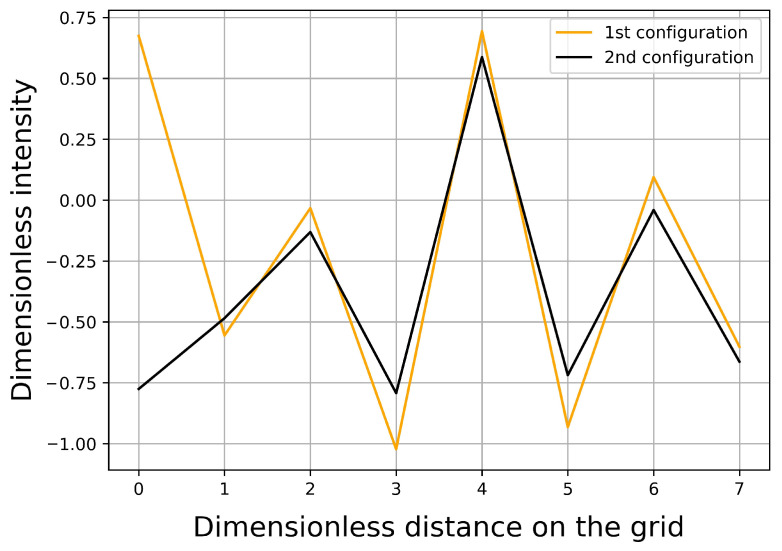
Similar characteristics of the scattered waves corresponding to the left-hand side cases.

**Figure 12 sensors-21-00011-f012:**
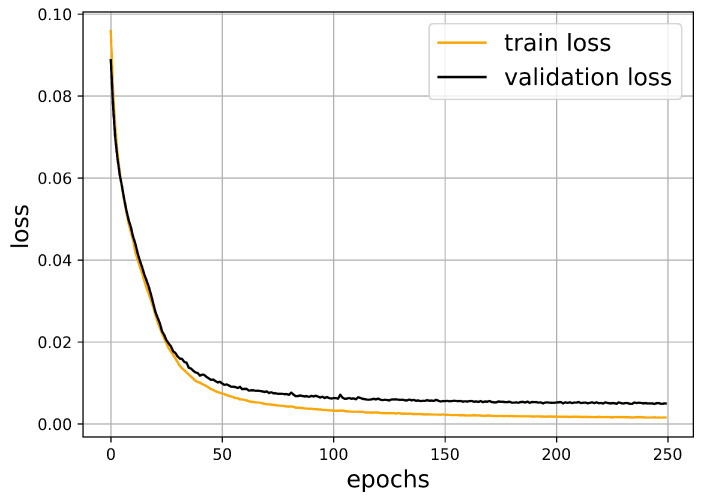
Train and validation loss for the unprocessed, raw simulation data with a locally connected layer.

**Figure 13 sensors-21-00011-f013:**
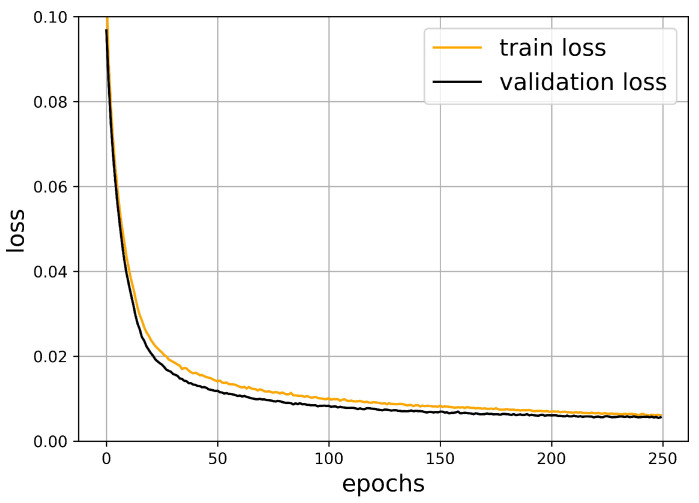
Train and validation loss for preprocessed simulation data with a conventional convolution layer.

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
