# Peer review of "Localization of Scattering Objects Using Neural Networks"

_sensors, 2020, doi:10.3390/s21010011_

Round 1
Reviewer 1 Report
An up-to-date approach: neural networks are used to estimate the localization of multiple scattering objects using scattered fields. In virtue of neural networks, using a relatively large training data set, the unknown locations of the objects can be estimated. However, the advantage of the proposed methods compared to the traditional ones are not clearly stated. So, the performance of the proposed methods with the other ones should be figured out in the numerical tests. Whether does the scattered fields used in the test included the noise? If not, please test the robustness of the proposed method. In figure 3, 11 data points of the original number 165 are used for the input of the neural networks. How did you determine the number of the date points? what is the principle? In figure 6 and 7, the symbols of ‘dev_b’, dist and ‘dev_g’ are not definied. Please defined these symbols clearly. In the training, how did you design the training samples? For example, how many objects are in the samples? please illustrate it. And according to the setting in the training, how is the generalization ability of the proposed methods?
Author Response
First of all, we are very grateful for the careful reading of the manuscript. The criticism and all of the remarks were justified and we have revised the manuscript according to these.
Point 1: ... However, the advantage of the proposed methods compared to the traditional ones are not clearly stated. So, the performance of the proposed methods with the other ones should be figured out in the numerical tests.
Response: Following the advice, we have displayed in the revised version the result of two further simulations. In the first one, we did not perform any preprocessing for the training data. In the second case, a conventional convolution layer is included instead of the more complex locally connected layers. In both cases, we obtained acceptable results, but they are still significantly less accurate compared to our approach. We have devoted the new subsection 3.2. to this issue; see the explanation here with Figure 12 and 13.
Point 2: Whether does the scattered fields used in the test included the noise? If not, please test the robustness of the proposed method.
Response: In the revised version, also noisy data were used in the simulations. We did not have to perform any pre-smoothing: it turned out that our preprocessing does this job automatically. This proved to be a lucky choice: the preprocessed noisy data is almost identical with the preprocessed "smooth" data. Therefore, the input of the neural network is almost identical in these two cases, such that we obtained the same accuracy as in the noise free simulations.
See lines 144-150, Figure 4 and the last five lines of Section 3.1. Moreover, in the supplementary materials, we have included a large noisy data set for testing purposes.
Point 3: In figure 3, 11 data points of the original number 165 are used for the input of the neural networks. How did you determine the number of the date points? what is the principle?
Response: It is widely accepted that for a reliable identification of a scattered wave, approximately 10 measurement-points are necessary per wave-length. This, in our simulations with frequency 4π on a length of 10 units would require 200 data points. Therefore, our choice can be recognized as a risky one. Since the whole procedure was working with this, we have used this number. Of course, this number could be further reduced using multi-frequency data, but this again would need a more complex network. In the new version, we have inserted an explanation in lines 113-117 regarding this issue.
Point 4: In figure 6 and 7, the symbols of ‘dev_b’, dist and ‘dev_g’ are not definied. Please defined these symbols clearly.
Response: In the new version, all of these variables are defined, see 248-250.
Point 5: In the training, how did you design the training samples? For example, how many objects are in the samples? please illustrate it. And according to the setting in the training, how is the generalization ability of the proposed methods?
In all cases, we have used two obstacles. Their midpoints were placed to the (different) grid points of the square [-4,4] x [-4,4]. Note we had to exlcude cases only with interchanging objects. This corresponds to the total number of 735 of our direct simulations. 92% of these (using a random choice) were used to train the network and 8% for test the performance. Two concrete cases are shown in Figure 1 and Figure 2. According to this comment, we have extended the first two paragraphs in Section 2.1.
Of course, one could generalize this to make the number of objects also an unknown parameter. For this, one should train the network with a number of samples for one object, with a number of samples for two objects, with a number of samples for three objects, etc. In this case, we would propose two networks. The first one with a discrete output should predict the number of objects and then a second network, similar to the presented one should perform the localization. This, however, is beyond te scope of our contribution. We only propose the principle for building the second "critical" neural network. The real difficulty in this topic is the presence of multiple objects, which could be solved (in a simple case) using our construction. We discuss these issues shortly in the last paragraph of the new Conclusion section.
According also to the poor scores to questions Is the research design appropriate? and Are the methods adequately described?, we note the following:
- The explanation in Section 2.2 was extended to add details on the locally connected layer as a main compound.
- The major part of Section 2.2.1 was also rewritten to make the description of the method more clear.
Reviewer 2 Report
Review report on the Research Article sensors-1011918 titled "Localization of scattering objects using neural networks", by Domonkos Haffner , Ferenc Izsák.
The paper deals with a basic 2D inverse scattering problem, which –in its general form- can be still considered as a challenging issue. The localization of a couple of “known-shaped” sound-hard disks is performed by a neural-network, which has been trained by a number of synthetic solutions of the well-posed direct problem (i.e., in the framework of supervised learning). The scattered filed measured on (only) one side of the square investigation domain is used to localize the centers of the disks.
This means that the unknown is represented by the two couple of coordinates (x_1,y_1) and (x_2,y_2) of the two disks 1 and 2, that is four real values.
In order to effectively use the measured data and to avoid overfitting, one of the key point of the proposal is the following: since the scattered data is very oscillating, it is first simplified by interpolation of some of its average values. This data simplification is useful in this paper, since the unknown (i.e., teo 2D positions) is very simple and the scattered data contains too much (and “unstable”) information. This is a sort of data regularization (as well as any high pass filtering on the scattered data).
The paper reports some numerical results, which has been computed by means of available packages (for the direct scattering problem and for the neural network).
The paper is enough well organized and correct, the general setting (i.e., inverse scattering) is interesting, and the numerical results are positive. Anyway, the problem statement is really too simple (the problem seems more suitable as exercise for a graduate student's thesis than an up-to-date research work). In general, a simple problem is important for a seminal paper, which gives a new idea or paradigm. This is not the case, because the paper is just a synthetic case study (i.e., the application of previous and well known methodologies to another –and simple- problem).
Please consider these major comments.
- Please improve a lot the first part of the Introduction (i.e., Secction 1 at page 1).
- Please give all the information about the choice of the convolution kernels and all the details (very precise) to reproduce the numerical results: this could be the main contribution of this paper, which could become intersting for the research comunity. Without that, the paper has a very low impact.
In addition, the are some minor comments to be considered.
- Please, write “total field” and “scattered field” in the Caption of Fig. 1 and 2 (not only at the top of the figures).
- Please explain the symbol Omega^- (instead of Omega).
- Please explain at the end of page 2 that Omega_0 is a side of the investigation domain (instead of explaining later on).
Author Response
We are very grateful for the careful reading of the manuscript and for the constructive comments.
Comment 1: Please improve a lot the first part of the Introduction (i.e., Secction 1 at page 1).
Response: We have extended this part significantly. We mentioned real-life cases where inverse scattering is of great importance and invesre scattering is beyond the analysis of scattered pulse-waves. We tackled also the problem of having incomplete observations, which is a rela practical limitation and corresponds to our geometrical settings.
Comment 2: Please give all the information about the choice of the convolution kernels and all the details (very precise) to reproduce the numerical results: this could be the main contribution of this paper, which could become intersting for the research comunity. Without that, the paper has a very low impact.
Response: We have completed the manuscript with these details, which can be found in Section 2.2 in the revised version. Following the advice, we have formulated everything in precise terms. Accordingly, we have also revised the first part of 2.2.1 to explain the adaptation of the above procedure to our network. A very detailed documentation is added to all the program codes, so the numerical values are easily reproducible.
Minor comment 1: Please, write “total field” and “scattered field” in the Caption of Fig. 1 and 2 (not only at the top of the figures).
Response: We have performed this modification.
Minor comment 2: Please explain the symbol Omega^- (instead of Omega).
Response: We have inserted a short explanation in line 55. The motivation for chosing this notation is hat in the theory of PDE's the domain, where the equations also solved, is usually denoted with Ω. We, however had to solve the scattering problem outside of the disks, and therefore we did not use the notation Ω for them.
Minor comment 3: Please explain at the end of page 2 that Omega_0 is a side of the investigation domain (instead of explaining later on).
Response: We did it now in line 65.
Further remarks:
According to the advice of the other reviewers, we have modified and completed the manuscript at varoius points. These are displayed with red color. Further minor grammar changes were also performed.
Reviewer 3 Report
The authors presented the concept of a simplified neural network for localization of scattering objects which is characterized by a complex locally connected layer. In general, the article is interesting, current, brings something new to the general state of knowledge but its application potential may raise doubts. After reading the content of the article, I have the following comments and questions:
- The research was based solely on simulation data. Such data is noise-free and therefore does not reflect the conditions of the real object. Model validation based on simulation data is insufficient. How will the authors respond to the objection that their solution has not been validated on real data, so there is no proof that it has any practical value?
- The authors argue that simplified shallow neural networks are as effective or even better than traditional deep networks. However, they did not carry out any comparative studies, including, for example, training a deep neural network using the same training data, confirming this thesis. In order to prove this thesis, the newly developed model should be compared with a deep neural network trained on the same data.
- The weakness of the research is that the authors deal with a very simple case. Reliable validation should cover many cases of varying number of inclusions, shape, size and location. What are the authors' arguments to counter the above allegations?
- The conclusions presented are too laconic and do not provide any relevant information on the results of the research. The Conclusions or Discussion section should discuss the results and how the results can be interpreted in perspective of previous studies and of the working hypotheses. The findings and their implications should be discussed in the broadest context possible and limitations of the work highlighted. Future research directions may also be mentioned. This section may be combined with Results.
- In general, this manuscript does not meet the requirements of the Sensors journal structure as set out here: https://www.mdpi.com/journal/sensors/instructions
- Figure 3 - both axes are not captioned.
- Figure 5 - the vertical axis is not captioned.
- Figure 6 - What do dev_b and dev_g mean?
- Figure 9 - both axes are not captioned.
- Figure 10 - both axes are not captioned. Check caption under this figure (y vs. y?).
- There is no description on how to run the attached scripts and supplementary data files.
Author Response
We are grateful for the careful reading of the manuscript. We found the comments to be justified and took all of them seriously to improve our work.
Comment 1: The research was based solely on simulation data. Such data is noise-free and therefore does not reflect the conditions of the real object. Model validation based on simulation data is insufficient. How will the authors respond to the objection that their solution has not been validated on real data, so there is no proof that it has any practical value?
Response: As proposed also by another reviewer, we have performed experiments also with noisy data. For this, we have added Gaussian noise to the simulated data set. The final result delivered the same accuracy as in the noise-free case. The main tool to achieve this was the preprocessing of the data. This not only reduces the number of parameters and prevents from overfitting but also minimizes of the effect of noise. Therefore, the input of the neural network is almost identical in these two cases, such that we obtained the same accuracy as in the noise free simulations.
See lines 144-150, Figure 4 and the last five lines of Section 3.1. Moreover, in the supplementary materials, we have included a large noisy data set for testing purposes.
This is still not a real data set. It is possible that in a practical case, one should fine-tune the proposed neural network. But what we did here was to fix the basis and principles of its construction.
Comment 2: The authors argue that simplified shallow neural networks are as effective or even better than traditional deep networks. However, they did not carry out any comparative studies, including, for example, training a deep neural network using the same training data, confirming this thesis. In order to prove this thesis, the newly developed model should be compared with a deep neural network trained on the same data.
Response: Our remarks on shallow networks reflects only the result of the reference [14], where the authors explain the details. We mentioned this research direction as an alternative of our approach. We, however, did not construct a shallow but a deep neural network as it has more hidden layers (see Figure 5).
In the revised version, we performed two new series of experiments comparing our method with the performance of two other neural networks. In this way, we pointed out the importance of of our main cornerstones, the preprocessing and the application of a complex locally connected layer. This is given in Section 3.2.
Moreover, we added the corresponding Python codes to the supplement material. These are commented and explained also in details.
Comment 3: The weakness of the research is that the authors deal with a very simple case. Reliable validation should cover many cases of varying number of inclusions, shape, size and location. What are the authors' arguments to counter the above allegations?
Response: Geometrically, this case is really simple. At the same time, the presence of multiple objects and in particular, the limitation in the observation (waves could detected only on the bottom edge of the square in Figure 1 and Figure 2) make this problem non-trivial; see also Figure 11. This limitation is rather realistic; think of seismic surveys or ground penetrating radar systems. We provided here a solid framework to handle this. To generalize our work for more objects, one should build first a network for detecting the number of objects and then the position and size could be forcasted. But for this, an enormous number of training data would be necessary.
Comment 4: The conclusions presented are too laconic and do not provide any relevant information on the results of the research. The Conclusions or Discussion section should discuss the results and how the results can be interpreted in perspective of previous studies and of the working hypotheses. The findings and their implications should be discussed in the broadest context possible and limitations of the work highlighted. Future research directions may also be mentioned. This section may be combined with Results.
Response: Following the advice, we have completed also Section 3 by discussing the results in more details and we have added Section 3.1 to discuss a comparison with other possible approaches. Also, the last section on conclusions was significantly extended and made more specific.
Comparing our study to the previous results, we related our results with the recent ones in [7] and [8]. In these, very complex shapes could be deteced but they assumed a "full" observation around the scattered object which is a single one and placed in a fixed location. This comparison was placed rather at the end of section together with the discussion of other references.
At the end of the new conclusion section, we mentioned the limitation of our work and the possibility to generalize our approach.
Comment 5: In general, this manuscript does not meet the requirements of the Sensors journal structure as set out here: https://www.mdpi.com/journal/sensors/instructions
Response: This was a real mistake. We had a new look to the requirements and fit the structure accordingly.
Comment 6: Figure 3 - both axes are not captioned.
Response: In the new version, the axes are captioned.
Comment 7: Figure 5 - the vertical axis is not captioned.
Response: In the new version, the vertical axis is captioned.
Comment 8: Figure 6 - What do dev_b and dev_g mean?
Response: In the new version, all of these variables are defined.
Comment 9: Figure 9 - both axes are not captioned.
Response: In the new version, the axes are captioned.
Comment 10: Figure 10 - both axes are not captioned. Check caption under this figure (y vs. y?).
Response: In the new version, the axes are captioned. The caption is correct now..
Comment 11: There is no description on how to run the attached scripts and supplementary data files.
Response: Detailed documentations are included now as follows:
- Basic informations are listed in the file readme.txt.
- At the beginning of each of the attached Python notebooks one can find a general description and instructions for running these.
- The Python notebooks are divided into blocks. They are also commented. In this way, each of the main steps can be run separately or changed.
- For the results, figures are generated and shown within the code.
Round 2
Reviewer 1 Report
The authors have well solved the revision and the paper can be accepted at current form.
Reviewer 2 Report
The paper has been improved in this second version. All the criticisms have been taken into account.
Please check carefully for typos in the manuscript, which can be do in the "proof" step of publication.
For instance "the scattered wave was detected the all around", and "R \times R^2" (instead of "R^2 \times R^2").
Reviewer 3 Report
After revision, the manuscript is ready for publication in its present form.